# The Role of the Skin Microbiome in Acne: Challenges and Future Therapeutic Opportunities

**DOI:** 10.3390/ijms252111422

**Published:** 2024-10-24

**Authors:** Alicja Niedźwiedzka, Maria Pia Micallef, Manuele Biazzo, Christine Podrini

**Affiliations:** The BioArte Ltd., Malta Life Science Park, Triq San Giljan, SGN 3000 San Gwann, Malta

**Keywords:** acne vulgaris, skin microbiome, *C. acnes*, microbiome-targeted therapies, probiotics

## Abstract

Acne vulgaris is a widespread dermatological condition that significantly affects the quality of life of adolescents and adults. Traditionally, acne pathogenesis has been linked to factors such as excess sebum production, follicular hyperkeratinization, and the presence of *Cutibacterium acnes* (*C. acnes*). However, recent studies have highlighted the role of the skin microbiome, shifting focus from individual pathogens to microbial community dynamics. This review critically evaluates existing research on the skin microbiome and its relationship to acne, focusing on microbial diversity, *C. acnes* strain variability, and emerging therapies targeting the microbiome. While certain studies associate *C. acnes* with acne severity, others show this bacterium’s presence in healthy skin, suggesting that strain-specific differences and overall microbial balance play crucial roles. Emerging therapeutic approaches, such as probiotics and bacteriophage therapy, aim to restore microbial equilibrium or selectively target pathogenic strains without disturbing the broader microbiome. However, the lack of standardized methodologies, limited longitudinal studies, and the narrow focus on bacterial communities are major limitations in current research. Future research should explore the broader skin microbiome, including fungi and viruses, use consistent methodologies, and focus on longitudinal studies to better understand microbial fluctuations over time. Addressing these gaps will enable the development of more effective microbiome-based treatments for acne. In conclusion, while microbiome-targeted therapies hold promise, further investigation is needed to validate their efficacy and safety, paving the way for innovative, personalized acne management strategies.

## 1. Introduction

The skin microbiota is the second largest microbial community in the human body after the gut microbiota. These microorganisms include bacteria, yeasts, archaea, viruses, and mites, all of which contribute to the skin’s overall health and function [1]. The adult skin microbiota is composed primarily of four major bacterial phyla: *Actinobacteria*, *Firmicutes*, *Proteobacteria*, and *Bacteroidetes*. Key genera include *Corynebacterium*, *Staphylococcus*, and *Cutibacterium*, with distribution varying by skin regions and conditions (e.g., dry, humid, or sebaceous/oily skin) [2].

Of particular interest is *Cutibacterium acnes* (*C. acnes*), a Gram-positive anaerobic bacterium that thrives in the lipid-rich, microaerobic environment of sebaceous glands [1]. *C. acnes* has long been considered a commensal bacterium; however, its involvement in various infections has led to its emergence as an opportunist pathogen. *C. acnes* is strongly linked to acne vulgaris (hereafter referred to as acne), a chronic inflammatory skin condition that primarily affects seborrheic (oily) areas such as the face, chest, and back [3]. Advances in metagenomic sequencing, especially 16S rRNA analysis, have deepened our understanding of *C. acnes* rybotipes and their roles in acne. For instance, the acne-associated ribotype RT4 thrives in sebaceous glands, forming biofilms and releasing virulence factors, while the non-acnegenic RT6 does not trigger inflammation [4].

Acne development is complex and involves several interconnected factors [5], as described in Table 1.

Conventional acne treatments, such as topical retinoids, benzoyl peroxide, and oral antibiotics, often target *C. acnes* and inflammatory pathways but can also disrupt the delicate balance of the skin microbiome [6]. This has fueled interest in microbiome-friendly treatments that restore microbial balance.

This review provides an updated overview of the skin microbiome’s role in acne pathogenesis, focusing on traditional and emerging treatments, antibiotic resistance risks, and the potential of microbiome-targeted therapies. Furthermore, we explore the potential role of probiotics and other microbiome-modulating approaches as promising alternatives to conventional acne treatments. Additionally, we highlight the role of opportunistic pathogens in skin infections, emphasizing how disruptions in skin microbiota can create an environment that allows these pathogens to proliferate, potentially exacerbating acne and other inflammatory conditions.

## 2. Commensal and Pathogenic Dynamics in the Skin Microbiota

Studies using sequencing techniques have shown that the composition of skin microbial communities in healthy adults largely depends on the specific characteristics of the skin site. Variations in microbial populations are linked to different skin environments—moist, dry, or sebaceous. Sebaceous areas, particularly the face and back, are primarily dominated by Cutibacterium species, which thrive in lipid-rich environments. In contrast, moist areas such as other body folds support a higher abundance of Staphylococcus and Corynebacterium species, which prefer humid conditions [1].

Fungal communities show less variability across facial and body sites compared to bacterial populations [5]. Fungi from the genus Malassezia dominate in core areas like the torso and arms, while non-sebaceous regions host a broader diversity of fungal species, including Aspergillus, Cryptococcus, Rhodotorula, and Epicoccum. Interestingly, bacteria far outnumber fungi across all skin sites, although this observation might partly be due to the larger number of available bacterial reference genomes [7].

Viruses, in contrast to bacteria and fungi, tend to colonize individuals in a more personalized manner, independent of the skin’s physiology, complicating the study of viral diversity. Advanced techniques, such as viral-like particle purification or metagenomic sequencing, are often necessary for studying these viruses due to the low biomass present in skin samples [8]. In fact, a study on human virome revealed that up to 94% of viral sequences from skin do not match known viral genomes, reflecting the limited data available in reference databases. The biggest part of the identified virome consisted of bacteriophages belonging to the Caudovirales order, with the majority predicted to be lysogenic [9]. Viral populations, particularly bacteriophages that infect *C. acnes*, play a role in modulating bacterial behavior. Bacteriophages can influence the balance between bacterial proliferation and host immune response, potentially affecting inflammation and lesion formation in acne [10]. In addition to bacteriophages, the skin harbors a variety of eukaryotic viruses, including human papillomaviruses and human polyomaviruses, as well as Merkel cell polyomaviruses that are present on the skin of healthy-appearing individuals [11]. Most studies focus mainly on the DNA virome, with the RNA virome of the skin still largely unexplored. These findings emphasize the need for further investigation into the dynamics and impact of the skin virome on human health.

Research using longitudinal sampling over two years has found that skin microbial communities are generally stable despite ongoing environmental changes [12]. This stability is largely due to the persistence of specific microbial strains rather than reacquisition from the environment. Sebaceous areas, in particular, show the greatest stability in both bacterial and fungal communities [13]. However, studies have shown that the use of cosmetics and even swimming in seawater [14] can temporarily change the composition of the skin microbiome. Interestingly, the application of cosmetics has been associated with a reduction in the relative abundance of typical skin bacterial groups. Simultaneously, non-core skin bacteria, such as Ralstonia, have been observed to increase in prevalence after cosmetic use, potentially due to their ability to metabolize certain cosmetic ingredients. While Ralstonia’s role in skin health is not fully understood, its rise indicates that cosmetics can shift the skin’s bacterial landscape [15]. Therefore, the impact of cosmetics on the skin microbiome should be considered in the context of both microbial diversity and skin health.

Dysbiosis refers to an imbalance in the composition of the microbiota, which plays a crucial role in modulating the immune system. When the skin’s microbial community is disrupted, it can lead to the absence of beneficial microbes and the proliferation of harmful ones, compromising the skin’s ability to maintain its normal functions and potentially triggering or worsening inflammatory conditions like acne [16]. Dysbiosis in the gut microbiota can contribute to skin dysbiosis through various mechanisms. For instance, gut dysbiosis can lead to a reduction in beneficial metabolites like short-chain fatty acids, which are essential in regulating the Th17 immune response as well as maintaining skin barrier integrity [17]. Additionally, gut dysbiosis may result in increased intestinal permeability, known as “leaky gut”, allowing toxins and pathogens to enter the bloodstream, thereby provoking systemic inflammation that affects the skin. This disruption can also shift the immune system toward a pro-inflammatory response, exacerbating acne [18]. Furthermore, the gut microbiota plays a role in hormone metabolism, and dysbiosis can disrupt hormonal balance, potentially influencing hormone-related skin conditions like acne [19]. While it remains unclear whether dysbiosis is a cause or consequence of certain skin diseases, recent findings suggest that it may play a primary role in initiating facial inflammation and lesion formation [20].

### 2.1. C. acnes and Its Influence on Acne Pathophysiology

Several non-infectious, inflammatory, and immunomodulatory roles of *C. acnes* in acne development have been identified. *C. acnes* can enhance local inflammation by stimulating sebocytes to release pro-inflammatory cytokines, such as tumor necrosis factor-α (TNFα), interleukin (IL)-6, IL-8, and IL-12 [21,22,23]. This occurs partly through Toll-like receptor 2 (TLR2) activation. Additionally, it induces IL-1β secretion in human monocytes via nucleotide oligomerization domain (NOD)-like receptor pathways, specifically through the NLRP3 inflammasome and caspase-1 activation [24]. *C. acnes* has also been shown to drive Th17/Th1 responses in T cells, promoting the production of IL-17A and interferon-γ (IFNγ) in vitro [25]. Components of Gram-positive bacterial cell walls, such as lipoteichoic acid and peptidoglycan, further stimulate keratinocytes to produce neutrophil-attracting cytokines like TNFα and IL-8 via TLR2 activation [26].

Beyond immune responses, *C. acnes* influences skin physiology by modulating keratinocyte differentiation, regulating lipid production in sebocytes, and triggering reactive adipogenesis in dermal fibroblasts [27]. Dermal fibroblasts are important cells that maintain the structural integrity of tissues. Reactive adipogenesis—where skin fibroblasts proliferate and differentiate into preadipocytes in response to bacterial stimuli—has been observed in human acne lesions, with *C. acnes* promoting this process through TLR2 in animal models [28]. Therefore, these fibroblasts are involved in the pathogenesis of acne and represent a potential target for acne therapy. Figure 1 represents *C. acne’s* influence on particular cell types, as summarized above.

*C. acnes* exists in six phylotypes: IA1, IA2, IB, IC, II, and III [29]. These phylotypes have been further classified into subgroups known as clonal complexes using multi-locus sequence typing and single-locus sequence typing (SLST) [30]. While *C. acnes* is present on both acne and non-acne skin, research has shown that acne-prone skin is associated with specific “acnegenic” phylotypes and a reduced diversity in *C. acnes* phylotypes. For instance, case–control studies highlighted a decrease in *C. acnes* phylotype diversity among patients with severe acne, showing a predominance of phylotype IA1 and SLST-type A1 [29]. The predominance of the IA1 phylotype, especially ribotypes such as RT4 and RT5, is strongly associated with inflammatory acne lesions [4]. These particular strains are more pathogenic due to their ability to form biofilms and produce pro-inflammatory enzymes, which exacerbate acne symptoms. In contrast, *C. acnes* ribotype RT6 is considered commensal and non-acnegenic, contributing to a balanced skin microbiome. However, another study comparing patients with mild and severe acne found that phylotype IA1 SLST type A1 was dominant in both groups [31]. Similarly, other research [20] did not detect significant differences in species diversity between healthy individuals and those with acne. Geographic variations may also play a role, as studies in Japan identified phylotype IA5 as strongly associated with acne, suggesting differences between Asian and European populations [32].

The interpretation of strain-level data can be challenging, as differences in results may stem from sampling techniques, which target different microbial niches, culture-based methods that may favor certain strains, or sequencing approaches like amplicon-based sequencing versus shotgun metagenomics, each of which introduces different biases.

### 2.2. The Role of Staphylococcus Species

Staphylococcus species are ubiquitous residents of human skin, playing a complex role as both potential pathogens and beneficial commensals. The skin is host to a diverse microbial community where Staphylococcus species, such as *S. epidermidis*, coexist with other microbes, contributing to skin health and disease. While these bacteria are generally harmless, their balance can be disrupted, leading to various skin conditions, including acne. A study by Dagnelie et al. highlighted the impact of dysbiosis on acne development [31]. The research demonstrated that an imbalance between *C. acnes* and *S. epidermidis* results in increased inflammatory responses. Specifically, dysbiotic conditions were associated with heightened inflammation compared to neutral microbial ratios. This suggests that maintaining a balanced microbial environment is crucial for preventing excessive inflammation and acne progression.

#### 2.2.1. Staphylococcus Capitis Strain E12 as *C. acnes* Inhibitor

Staphylococcus capitis (*S. capitis*) strain E12 exhibits significant antimicrobial properties, particularly against *C. acnes* [33]. This strain inhibits the growth of *C. acnes* more effectively than traditional antibiotics. The antimicrobial activity of *S. capitis* E12 is attributed to the production of phenol-soluble modulins (PSMs), which have selective antimicrobial effects. PSMs target and disrupt the membranes of pathogenic bacteria, thereby inhibiting their growth while preserving beneficial commensals.

Experimental studies involving pig skin and mice models have demonstrated the efficacy of *S. capitis* E12 in selectively targeting *C. acnes* without adversely affecting other beneficial skin bacteria [34]. The selective action of antimicrobial peptides produced by *S. capitis* E12 underscores its potential as an alternative therapeutic agent for acne treatment.

#### 2.2.2. Staphylococcal-Produced Bacteriocins and Their Selective Inhibition of Pathogens

A part of *Staphylococcus* species naturally occurring on skin, including *S. epidermidis*, produces antimicrobial peptides called bacteriocins. These peptides exhibit selective inhibition of opportunistic pathogens [33,34]. The specificity of bacteriocins towards harmful bacteria while sparing beneficial microbes makes them valuable in maintaining a balanced skin microbiome and preventing infections. A recent study [35] indicates that bacteriocins disrupt bacterial cell membranes and interfere with cell wall biosynthesis, providing a targeted approach to inhibit pathogens without promoting antibiotic resistance. The study emphasizes the potential of bacteriocins, particularly AS-48, as candidates for treating infectious diseases. AS-48 showed potent activity against clinical *C. acnes* isolates, selectively binding to bacterial membranes while sparing eukaryotic cells. Moreover, its combination with lysozyme enhances its bactericidal effect and reduces minimum inhibitory concentrations, highlighting AS-48’s promise as an effective treatment for dermatological infections, especially those involving biofilm-forming pathogens.

#### 2.2.3. The Potential of Staphylococcus in Acne Treatment

Understanding the role of the Staphylococcus species in acne pathogenesis opens new avenues for therapeutic interventions. By leveraging the antimicrobial properties of *S. capitis* strains and bacteriocins, researchers can develop targeted treatments that address both the pathogenic aspects of acne and the need to maintain a healthy skin microbiome. These novel treatments could include topical applications of antimicrobial peptides or formulations that modulate microbial balance to prevent acne development whilst reducing the heavy reliance on traditional antibiotics.

The increasing threat of antimicrobial resistance has spurred interest in the protective role of skin bacteria and their competition against pathogens. A survey of cultivable bacteria from human skin identified 21 bacteriocins capable of inhibiting various Gram-positive bacteria, including *Staphylococcus epidermidis*, methicillin-resistant *Staphylococcus aureus* (MRSA), and, most importantly, *C. acnes*. The majority of these antimicrobial-producing bacteria were found to be strains of the *Staphylococcus* genus [36]. These findings demonstrate the antimicrobial potential that could be harnessed from within the human skin microbiota, especially at a time when antibiotic resistance is of major concern.

In conclusion, the intricate interactions between Staphylococcus species and other skin bacteria play a critical role in acne pathogenesis. Advances in understanding these interactions provide promising strategies for developing new treatments that target microbial imbalances and inflammatory processes associated with acne.

### 2.3. The Malassezia Genus in Skin and Malassezia-Associated Skin Diseases

Although bacteria make up the majority of the skin microbiome, fungi are also widespread. Malassezia is the most abundant genus in the fungal microbiota found on human skin. Malassezia globosa, Malassezia restricta, and Malassezia sympodialis are most commonly isolated from healthy human skin, especially regions rich in sebum [7]. Malassezia may not only survive on our skin to utilize the lipid-rich environment but could also offer protection against pathogenic microbes like *S. aureus* [37]. However, despite being part of the normal human cutaneous microbiota, Malassezia yeasts have been linked to a number of skin disorders such as dandruff, pityriasis versicolor, seborrheic dermatitis (SD), Malassezia folliculitis (MF), and atopic dermatitis (AD) [38]. This suggests that they can exist both as commensal organisms and as pathogens, making their interactions with the human immune system particularly interesting.

The reason why Malassezia is so concentrated in sebaceous regions is because of their inability to synthesize fatty acids; therefore, they rely on external sources of fatty acids for their nutritional needs. Human sebum contains triglycerides, wax monoesters, squalene, cholesterol, and cholesterol esters produced by the sebaceous glands [39]. Malassezia spp. that reside on the skin release enzymes, most importantly, lipases, which result in the degradation of sebum to free fatty acids, which are then taken up to carry out a variety of important biological processes [40].

As a skin commensal, Malassezia first interacts with the immune system through the skin, which is actively involved in the body’s immune defenses. In healthy skin, Malassezia is known to interact with keratinocytes and immune cells. Its cell wall components, mainly β-(1,6)-glucans, glycolipids, and glycoproteins, are detected by proline-rich region (PRR) motifs present on specific receptors, primarily Dectin-2, Mincle, and Langerin, in multiple immune cell types [41,42]. However, the exact roles these receptors play in regulating Malassezia-induced commensalism and immune responses are still not fully understood and require more research. Studies have shown how M. furfur, M. globosa, and M. restricta stimulate the production of pro-inflammatory cytokines, chemokines, and antimicrobial peptides in keratinocytes. This includes increased expression of TLR-2, IL-8, and human beta-defensins 2 and 3 [43]. The majority of investigations relating to the host-fungal interplay are conducted with isolated host cells in vitro, which may not accurately reflect in vivo conditions. An experimental model using mouse skin demonstrated how Malassezia can activate Th17 immunity through the IL-23/17 axis; more specifically, the authors identified a CCR6+ Th17 subset of memory T cells that is specific to Malassezia in both healthy individuals and patients with atopic dermatitis, with the latter exhibiting a higher frequency of these cells [44]. The mechanisms underlying Malassezia-associated skin conditions remain unclear due to the limited understanding of how Malassezia facilitates an immune response for either a commensal or inflammatory state in human skin.

Similar to Cutibacterium, the lipophilic characteristics of Malassezia spp. highlight the potential influence of this fungus on acne. Apart from *C. acnes*, Malassezia spp. have also been isolated from acne lesions, and therefore, it has been suggested that Malassezia can also induce acne [45,46,47]. Additionally, it has been shown that the lipase activity of Malassezia is more than 100 times stronger than that of *C. acnes* [48], suggesting that Malassezia may play a more direct role in acne pathogenesis than previously suggested. A study by Hu et al. [49] demonstrated the effectiveness of antifungal treatment in patients with refractory papules and pustules. Acne lesions were significantly reduced, and, in some cases, they disappeared after discontinuation of antibiotics with combined application of antifungal treatments, leading the authors to propose that Malassezia, not *C. acnes*, may be a potential cause of refractory acne. The findings above warrant further insight into the role of Malassezia spp. on acne. The causal link between Malassezia and skin conditions like seborrheic dermatitis and dandruff has been demonstrated through several findings: (1) successful treatment of these disorders with antifungal medications, (2) symptom improvement accompanied by a quantitative reduction in Malassezia, and (3) dandruff recurrence triggered by Malassezia metabolites [50]. However, there is not yet enough evidence to suggest a definitive causal relationship between Malassezia and acne.

## 3. Antimicrobial Resistance in Acne: Mechanisms, Complications and Recommendations

Topical antibiotics such as clindamycin and erythromycin are frequently used to combat *C. acnes* and other bacteria involved in acne. These antibiotics inhibit bacterial protein synthesis, reducing the bacteria that contribute to acne. However, their broad-spectrum activity can disrupt the skin’s microbiome by affecting both harmful and beneficial bacteria. The emergence of *C. acnes* resistance has been linked to the introduction of topical antibiotic treatments in the 1970s [51]. Clinically significant resistance of *C. acnes* to antibiotics was first documented in 1983 in the U.S. among patients who did not respond well to oral antibiotic therapy [52]. Over time, various antibiotics, including topical clindamycin, erythromycin, oral macrolides, tetracyclines, and cephalexin, have been extensively used in acne management [53]. Reports of antimicrobial resistance in other bacteria from acne patients, such as *S. aureus* and *S. epidermidis*, have also been noted. A study involving Korean acne patients reported high resistance rates of *S. epidermidis* to tetracycline (31%), doxycycline (27%), clindamycin (33%), and erythromycin (58%) [54]. Additionally, a related Indonesian study involving 93 isolates showed 42.9% susceptibility to erythromycin and 71.4% to tetracycline [55]. For instance, a study in Jordan found that 35% of *S. aureus* and 25% of *S. epidermidis* isolates were antibiotic-resistant [56]. These findings highlight the growing concern of antibiotic resistance in acne treatment, emphasizing the need for alternative therapeutic approaches and more judicious use of antibiotics to preserve their efficacy.

### 3.1. Clinical Implications of Antibiotic Resistance in Acne

A French study involving 1472 hospitalized patients showed that all Cutibacterium strains were susceptible to amoxicillin, ceftriaxone, vancomycin, and moxifloxacin [57]. However, 15% of *C. acnes* strains exhibited resistance to erythromycin, 4.1% to clindamycin, and 2.2% to tetracycline. This resistance highlights the adaptive nature of *C. acnes*, especially when it forms biofilm communities on the skin and within sebaceous glands [58]. In biofilms, *C. acnes* is shielded from antibiotics, as the biofilm matrix reduces the penetration of effective drug concentrations [59]. Within these biofilms, the bacteria communicate through quorum sensing and exhibit reduced metabolic activity, which further protects them from antibiotic treatments [60]. When *C. acnes* shifts from the biofilm to a planktonic state, however, it can trigger new protein expression, enhancing its ability to tolerate antibiotics. This dual behavior of *C. acnes*—biofilm formation and planktonic adaptation—demonstrates the complexity of its role in both antibiotic resistance and maintaining balance in the skin microbiome. Studies have shown varying effects of *C. acnes* on biofilm development in other bacteria. For example, *C. acnes* supernatants reduced *S. aureus* biofilm mass in some cases, increasing antibiotic susceptibility [61]. Furthermore, *C. acnes* biofilm can be degraded by an enzyme secreted by Cutibacterium granulosum, emphasizing the balance between microbial species in preventing pathogen colonization [62]. These findings stress the need to maintain microbiome balance when using antibiotics. Although biofilm is considered a primary factor that ensures *C. acnes* persistence during acne antibiotic treatment, factors that promote early bacteria adhesion and biofilm formation have not been identified yet [63].

The global rise in antibiotic-resistant *C. acnes* poses a growing challenge in acne management, with varying resistance patterns across different regions. Factors such as diverse prescribing practices, concomitant use of topical treatments, and differences in *C. acnes* populations likely contribute to these discrepancies. The emergence of resistant bacterial strains not only diminishes the effectiveness of antibiotic therapies but also raises concerns about promoting resistance in other skin microbiota. This highlights the need for alternative treatment strategies that minimize antibiotic use while maintaining the balance of the skin microbiome.

Earlier studies raised concerns about treatment failure in acne due to antibiotic-resistant *C. acnes*. However, these studies largely involved antibiotic monotherapy, which is now discouraged. Current practice recommends using topical antibiotics in combination with benzoyl peroxide (BPO), which enhances efficacy and reduces resistance [64]. BPO has anti-Cutibacterium effects regardless of antibiotic resistance, synergizes with topical antibiotics, and can reverse resistance in certain strains [65]. Additionally, oral tetracyclines are effective not only for their antibacterial properties but also for their anti-inflammatory effects. To reduce the risk of resistance, the use of antibiotics for acne should be limited to less than 12 weeks, monotherapy avoided, and non-antibiotic treatments favored [64].

*C. acnes* is also implicated in other conditions, including sarcoidosis [66], prostate cancer [67], and post-surgical infections [68]. The role of *C. acnes* in biofilm formation on medical implants, such as shoulder arthroplasty, has led to the exploration of non-antibiotic approaches for infection prevention. For example, a study demonstrated that benzoyl peroxide, when applied preoperatively, significantly reduced *C. acnes* in shoulder surgeries, preventing infection [69]. Overuse of antibiotics in acne management can also promote resistance in other bacterial species, posing risks to treatment efficacy for other infections. A study involving systemic doxycycline found increased resistance in staphylococcal skin colonizers after treatment [70]. Another study reported changes in skin microbiota composition after oral minocycline treatment, with increases in Pseudomonas and Streptococcus species and reductions in beneficial Lactobacillus species [71].

Antibiotic overuse can lead to resistant Staphylococci on the skin, which may spread to untreated family members. Similarly, dermatologists specializing in acne treatment were found to be colonized with resistant Cutibacterium species [72]. Beyond the skin, antibiotic use for acne may increase the risk of respiratory and urinary tract infections. For instance, a UK study reported a two-fold increase in upper respiratory tract infections among acne patients on antibiotics [73].

There is also emerging concern about the impact of acne antibiotics on the gut microbiome, which is increasingly recognized for its role in non-infectious diseases. In vitro studies showed significant disruptions in gut microbiota after exposure to acne antibiotics, raising concerns about long-term health effects [74]. A study comparing the effects of anti-acne antibiotics sarecycline, minocycline, and doxycycline on the gut microbiome showed that sarecycline caused the least disruption, with most bacterial populations recovering after treatment and minimal impact on key species such as Lactobacillus, Enterobacteriaceae, and Bacteroides. In contrast, minocycline led to the most significant dysbiosis, reducing beneficial bacteria like Lactobacillus and causing the loss of important families such as Ruminococcaceae and Clostridiaceae, with incomplete recovery. Doxycycline also reduced microbial diversity, notably decreasing Lactobacillus and Bacteroides populations and promoting overgrowth of Enterococcus species, resulting in a slower recovery of gut microbiota [74].

### 3.2. Prescribing Patterns and Guidelines for Acne Treatment

Antimicrobial resistance concerns have influenced current acne treatment guidelines. European and U.S. guidelines [6] recommend limiting the use of systemic antibiotics to doxycycline and lymecycline for no more than three months. Monotherapy with topical antibiotics is discouraged, and the use of oral antibiotics other than tetracyclines and macrolides is generally not recommended due to limited efficacy data. In recent years, sarecycline, a narrow-spectrum tetracycline derivative, has been approved for acne treatment in the U.S., and the topical use of broad-spectrum antibiotics like nadifloxacin has been questioned due to resistance risks.

Despite awareness of antibiotic resistance, World Health Organization reports show that antibiotic prescribing practices remain variable [75]. For instance, a study found that two-thirds of acne patients received antibiotic courses exceeding six months [76]. Dermatologists and non-dermatologists alike have prescribed prolonged antibiotic therapy for acne, underscoring the need for greater awareness and adherence to guidelines.

Resistance in *C. acnes* is primarily due to point mutations in ribosomal RNA subunits, with erythromycin and tetracycline resistance documented globally. European and U.K. studies show that resistance rates vary between countries, reflecting differences in antibiotic prescribing practices [77]. Cross-resistance between macrolides and clindamycin has been observed due to mutations in ribosomal RNA genes [78]. Resistance to tetracycline is typically associated with mutations in the small ribosomal subunit [79].

### 3.3. Recommendations and Future Directions

To mitigate the risk of antimicrobial resistance, current guidelines recommend using antibiotics judiciously, in combination with benzoyl peroxide, and for limited durations [64]. Non-antibiotic treatments should be prioritized, and physicians should be educated on the risks of resistance. Addressing antibiotic misuse is essential for maintaining effective treatment options and preventing broader public health implications.

## 4. Impact of Approved Non-Antibiotic Acne Treatments on the Skin Microbiome

Acne is a multifaceted condition treated through a variety of approaches, as recommended by the European and UK National Institute for Health and Care Excellence [6,80]. The current guidelines encompass both topical and systemic therapies, each designed to address specific aspects of acne pathophysiology, such as inflammation, bacterial overgrowth, and sebum production. Understanding the influence of these treatments on the skin microbiome is crucial for optimizing therapeutic strategies while minimizing potential disruptions to skin health.

### 4.1. Topical Treatments

Topical therapies play a central role in managing acne, with options including azelaic acid, BPO, and retinoids [81]. Azelaic acid (AA) is a commonly used treatment due to its moderate antimicrobial properties and effectiveness in reducing acne lesions. However, its impact on *C. acnes* is limited compared to other agents. This highlights the need for adjunctive treatments in more severe cases. AA is also effective for rosacea and melasma, often showing equivalent results to other treatments. It can be particularly useful when patients cannot tolerate other options or when certain therapies, like retinoids, are contraindicated, such as during pregnancy. While generally safe, potential side effects include itching and pain [82].

BPO stands out for its potent bactericidal action against *C. acnes*. It effectively reduces microbial diversity and lowers *C. acnes* prevalence, demonstrating its strong efficacy in acne management. BPO does not contribute to antibiotic resistance, making it a valuable option, particularly when used in conjunction with antibiotics or as a standalone treatment for short durations [83]. The use of BPO can help mitigate the development of resistant bacterial strains, which is a significant concern with prolonged antibiotic use.

Topical retinoids, including adapalene, tretinoin, and isotretinoin, are effective in treating comedonal acne by promoting cell turnover and preventing the formation of comedones. While retinoids do not possess direct antimicrobial effects, they can indirectly influence the microbiome by reducing sebum production [84,85,86]. They are generally well tolerated and show comparable efficacy to BPO and AA, although individual responses can vary. For patients with darker skin tones, hyperpigmentation is one of the most prevalent and bothersome dermatologic conditions, often caused by acne. Early and aggressive treatment of acne is essential to prevent hyperpigmentation. Retinoids can help improve hyperpigmentation due to their anti-inflammatory properties. For patients with sensitive skin, newer retinoid formulations in lotions may be less irritating than older ones [87].

### 4.2. Systemic Treatments

Oral isotretinoin remains the gold standard for severe acne, particularly in cases of nodular or conglobate forms. Isotretinoin works by reducing sebum production and indirectly affecting the microbiome through a reduction in *C. acnes* density [88]. Research indicates that isotretinoin increases microbial diversity and correlates with improved clinical outcomes [89]. Despite its efficacy, isotretinoin requires careful monitoring due to its potential side effects, which include mucocutaneous dryness and potential teratogenic effects.

Hormonal anti-androgens provide an alternative acne treatment for women, particularly those with elevated androgen levels, by targeting androgens like testosterone that stimulate sebum production. Research has shown that increased levels of androgens such as testosterone, androstenedione, and dehydroepiandrosterone correlate with acne severity in many women, especially those with conditions like polycystic ovary syndrome [90]. However, acne severity is not always directly linked to androgen levels, as other factors like sex hormone-binding globulin and neuroendocrine controls also play significant roles. Hormonal therapies, including contraceptives and anti-androgens, can reduce acne severity by increasing sex hormone-binding globulin levels and lowering the bioavailability of active testosterone [91]. Despite mixed findings on the correlation between androgen levels and acne, hormonal anti-androgens remain effective for many women, particularly when other treatments fail.

## 5. Emerging Non-Antibiotic Medications for Acne

### 5.1. Trifarotene

One of the most recent non-antibiotic treatments approved for acne is trifarotene (Aklief^®^), a fourth-generation selective retinoid [92]. Unlike earlier retinoids, trifarotene specifically targets the retinoic acid receptor gamma (RARγ), offering more precise modulation of acne-related processes like follicular differentiation and inflammation. Approved in 2019, trifarotene has shown significant efficacy in clinical trials, demonstrating a reduction in both inflammatory and non-inflammatory acne lesions on the face and trunk within as little as four weeks. Over a 52-week study, patients continued to see improvements in skin health and quality of life [93]. Trifarotene’s localized action and minimal systemic absorption make it a promising non-antibiotic option for acne management, especially as concerns over antibiotic resistance grow.

In addition to its acne-reducing effects, trifarotene has shown promising results in treating acne-induced hyperpigmentation across all skin types. A phase IV double-blind study [94] evaluated trifarotene combined with a skincare regimen (moisturizer, cleanser, and sunscreen) in patients aged 13–35 with moderate acne vulgaris and hyperpigmentation. Over 24 weeks, trifarotene significantly improved disease severity and hyperpigmentation scores compared to the vehicle. Patients treated with trifarotene experienced greater reductions in both hyperpigmentation and acne lesions, along with higher treatment satisfaction and fewer adverse events, highlighting the benefits of combining trifarotene with a skincare routine for comprehensive acne care [95].

### 5.2. Clascoterone

Clascoterone represents a novel and emerging approach in the treatment of acne vulgaris, offering a topical alternative to systemic anti-androgen therapies. As the first-in-class steroidal anti-androgen used topically, clascoterone specifically targets androgen receptors in the skin, blocking the effects of dihydrotestosterone (DHT) and thereby reducing sebum production and inflammation—two key drivers of acne [96]. Its rapid local metabolism to an inactive form minimizes systemic exposure and associated side effects, making it a safer option. Clascoterone’s ability to modulate multiple acne-causing pathways positions it as a promising addition to the evolving landscape of acne management.

Two Phase 3 trials assessed the safety and efficacy of clascoterone cream 1% for moderate to severe acne in patients aged 9 and older. Over 12 weeks, clascoterone showed significantly higher success rates and reductions in both inflammatory and non-inflammatory lesions compared to the vehicle cream [97]. It was well tolerated, with primarily mild adverse events. A long-term extension study further demonstrated that treatment success and lesion reduction continued to improve with prolonged use, confirming clascoterone’s efficacy as a topical therapy for acne [98].

### 5.3. Oral Spironolactone

Spironolactone, a synthetic steroid, is used off-label for acne treatment due to its anti-androgen properties. By blocking dihydrotestosterone from binding to androgen receptors in sebocytes, it reduces sebum production, making it a potential alternative to oral isotretinoin and hormonal contraceptives for long-term acne management in women. Although primarily a diuretic, spironolactone’s acne benefits have made it a valuable option, especially for women with premenstrual acne flares, offering an alternative to antibiotics without contributing to antimicrobial resistance [99].

Clinical studies have demonstrated the efficacy of spironolactone in treating acne in adult women. A pragmatic, phase 3 randomized controlled trial evaluated spironolactone in women aged 18 and older with persistent facial acne [100]. Participants receiving spironolactone showed greater improvement in acne-specific quality of life scores compared to those on placebo, with more pronounced results at 24 weeks. Treatment success, based on the investigator’s global assessment, was significantly higher in the spironolactone group, with no serious adverse reactions reported. Mild side effects, such as headaches, were slightly more common with spironolactone.

In a similar randomized superiority trial, spironolactone was again shown to be more effective than a placebo in reducing acne symptoms and improving quality of life [101]. Women treated with spironolactone were more likely to report overall improvement in acne by week 24, with investigator-reported treatment success also favoring the spironolactone group. Adverse effects were minimal, with no serious events recorded. Together, these studies confirm spironolactone’s efficacy and safety in managing acne in women, offering a promising non-antibiotic treatment option.

### 5.4. NAC-GED Gel

A novel therapeutic approach using NAC-GED, a PPARγ modulator, has shown promising results in the treatment of moderate-to-severe facial acne vulgaris. In a double-blind, phase II randomized controlled trial conducted across multiple sites in Germany, Italy, and Poland, NAC-GED gel (at concentrations of 5% and 2%) demonstrated significant efficacy in reducing total lesion count and achieving Investigator Global Assessment success compared to a placebo [102]. The 5% concentration, in particular, proved to be more effective and statistically superior to the vehicle group. The 2% concentration also showed notable improvements, although it did not achieve statistical significance in success compared to the placebo. The treatment was well tolerated, with adverse event rates comparable to the vehicle group. These findings suggest that NAC-GED 5% gel is a promising first-in-class PPARγ modulator for acne treatment, warranting further investigation in phase III clinical trials to confirm its efficacy and safety.

A cost-effectiveness analysis of NAC-GED 5% compared to BPO plus adapalene for moderate-to-severe acne vulgaris within the UK’s National Healthcare Services highlighted its potential as a valuable treatment. The study, informed by NICE Guidelines, found NAC-GED 5% economically justifiable. It demonstrated comparable efficacy to oral isotretinoin with a better safety profile, including lower discontinuation rates and a faster onset of action [103]. These findings suggest NAC-GED 5% as a cost-effective alternative in acne management, benefiting both patients and healthcare systems.

## 6. Bioactive Approaches to Optimize Microbial Balance

### 6.1. Probiotics

Probiotics are live microorganisms that confer health benefits and are increasingly recognized for their potential in acne treatment. While traditionally used in oral and dietary applications, emerging research highlights their topical and oral use in dermatology and cosmetology. Key probiotic strains, such as *Lactobacillus* and *Bifidobacterium*, exhibit protective effects on skin health, promoting beneficial skin microflora and enhancing the skin’s immune response [104]. Recent studies have demonstrated that topical probiotics can improve conditions like acne by increasing ceramide production, reducing inflammation, and acting as a barrier against harmful pathogens.

A study evaluated the efficacy and safety of a probiotic-derived lotion made from *Lactocaseibacillus paracasei* compared to 2.5% benzoyl peroxide for treating mild-to-moderate acne vulgaris. Both treatments significantly reduced inflammatory acne lesions and erythema index from baseline, with no statistically significant difference between the groups. However, comedones were not affected by either treatment. Side effects were reported in a smaller proportion of patients using the probiotic lotion compared to those using benzoyl peroxide. The findings suggest that the probiotic-derived lotion is a safe and effective alternative to benzoyl peroxide, with fewer side effects [105].

Building on these promising findings, the exploration of probiotics like *Lactiplantibacillus plantarum* (*L. plantarum)* offers an innovative approach to acne management. Topical probiotic formulations such as SkinDuo™ containing *L plantarum* leverage the beneficial effects of live probiotics to enhance skin health, demonstrating significant efficacy in reducing the proliferation of acne-causing bacteria, including *C. acnes* and *S. epidermidis*. Moreover, research on the in vitro topical application of *L. plantarum* has shown a reduction in the production of pro-inflammatory mediators like IL-1α, IL-6, and IL-8, which exacerbate acne lesions [106]. By addressing the root causes of acne through the modulation of skin microbiota and the inflammatory response, probiotics present a promising alternative or adjunct to conventional acne treatments. This novel strategy not only targets the symptoms of acne but also aims to restore and maintain the skin’s natural microbiome balance, positioning probiotics as a compelling area.

Additionally, oral probiotics have shown promise in reducing the side effects of antibiotic treatments while providing a synergistic anti-inflammatory effect. Recent studies highlight *L. plantarum* as a potential candidate for acne treatment. A placebo-controlled trial [107] revealed that supplementation with probiotics significantly improved acne severity, with reductions in inflammatory and total lesion counts. Participants also experienced enhanced skin hydration, decreased sebum triglyceride levels, and increased ceramide levels, suggesting improved skin barrier function. Moreover, the probiotic treatment shifted urine microbial profiles favorably, indicating that *L. plantarum* may effectively modulate inflammatory pathways linked to acne and serve as an adjunct or alternative to conventional therapies.

Another potential probiotic species could be *Lacticaseibacillus rhamnosus* (*L. rhamnosus*), which has shown effectiveness in acne vulgaris treatment. A clinical trial [108] demonstrated that those taking a probiotic capsule containing *L. rhamnosus* and *Arthrospira platensis* experienced significant improvement in acne severity compared to the placebo group. Notably, a greater percentage of patients in the probiotic group achieved a reduction in both total and non-inflammatory acne lesions, and there was a higher rate of improvement according to both the Acne Global Severity Scale and the Global Acne Grading System. Importantly, the probiotic treatment was well tolerated, with similar adverse events reported between groups.

### 6.2. Prebiotics

Prebiotics are non-digestible compounds that selectively stimulate the growth and activity of beneficial microorganisms. In the context of acne, prebiotics can enhance the efficacy of probiotics by providing essential nutrients that promote the proliferation of beneficial bacteria on the skin, which helps balance the skin’s microbiome. Commonly used prebiotics include alpha-glucan oligosaccharide, beta-glucan, fructooligosaccharides (FOS), galactooligosaccharides (GOS), and inulin. These compounds are known to support the growth of beneficial skin microbiota while helping to maintain a balanced skin environment, which can be particularly beneficial in conditions like acne [109].

Although prebiotics are less studied than probiotics, emerging research suggests their potential role in acne management. A study investigating the effects of FOS and GOS supplementation in women with adult female acne demonstrated improvements in blood parameters related to sugar and lipid metabolism. These prebiotics were associated with enhanced insulin sensitivity and reduced serum glucose and total cholesterol levels over a three-month period. Additionally, FOS and GOS supplementation promoted the growth of Bifidobacterium and Lactobacilli, which are essential for maintaining an efficient intestinal mucosal barrier. By improving gut health, prebiotics may reduce systemic inflammation, oxidative stress, and insulin resistance, all of which contribute to acne pathogenesis [110].

Further research is needed to fully understand the mechanisms by which prebiotics influence skin health, but these findings suggest that prebiotics, alongside probiotics, could be a valuable tool in acne management.

### 6.3. Synbiotics

The combination of probiotics and prebiotics, called synbiotics, offers a synergistic approach to acne management by enhancing the survival and activity of beneficial microorganisms while also providing essential nutrients to support their growth. This dual-action strategy has shown the potential to improve overall skin health and reduce acne severity [111].

A recent study investigated the efficacy of a dietary supplement containing *Bifidobacterium breve*, *Lacticaseibacillus casei*, *Ligilactobacillus salivarius*, and botanical extracts in subjects with mild to moderate acne. The authors demonstrated significant improvements in reducing superficial inflammatory acne lesions, sebum secretion, and skin desquamation. These effects were most pronounced in the group receiving both probiotics and botanical extracts, highlighting the potential of synbiotics to enhance acne treatment outcomes. Moreover, the treatment led to a reduction in *C. acnes* and *S. aureus*, while the beneficial *S. epidermidis* increased. This shift in microbial abundance suggests that synbiotics may help restore a healthy skin microbiome, reducing inflammation and supporting the skin’s natural defense mechanisms [112]. The findings indicate that synbiotics, through their combined probiotic and prebiotic action, can provide a safe and effective complement to traditional acne treatments, targeting both the symptoms and underlying dysbiosis linked to acne pathogenesis.

## 7. Phage-Therapy

Phage therapy represents a novel and promising approach to treating acne, particularly in the context of rising antibiotic resistance. Bacteriophages, or phages, are viruses that specifically target bacteria, making them an attractive option for reducing pathogenic bacterial populations without disrupting beneficial microbes. Unlike broad-spectrum antibiotics, which can indiscriminately kill both harmful and beneficial bacteria, phages offer the advantage of bacterial specificity. This specificity could potentially reduce the side effects associated with traditional acne treatments [113].

Recent studies have highlighted the potential of phage therapy in managing *C. acnes*. Notably, research has shown that acne patients often exhibit a reduction in *C. acnes* phage abundance compared to healthy individuals [114]. This reduction in phage levels correlates with an overgrowth of *C. acnes*, contributing to the inflammatory lesions characteristic of acne. By replenishing these phages, it may be possible to restore balance to the skin microbiome and reduce acne severity.

Preclinical studies using animal models have provided encouraging results for phage therapy in treating *C. acnes*-induced acne-like lesions. In one study, eight newly isolated phages were tested against antibiotic-resistant *C. acnes* strains. When used in mice, phage treatment significantly reduced both bacterial load and inflammation in the induced lesions. These findings highlight the potential of phage therapy as a complementary tool to antibiotics, particularly in combating antibiotic-resistant strains [115].

Human trials have also begun to explore the efficacy and safety of phage-based treatments. One study formulated a phage cocktail into a topical gel, which was tested on individuals with mild-to-moderate acne [116]. The high-dose formulation led to a significant reduction in *C. acnes* bacterial load, indicating that phage therapy could potentially reduce acne-causing bacteria in a dose-dependent manner. Although this study did not assess the impact on clinical acne severity, the results are promising and warrant further investigation.

Overall, while phage therapy for acne is still in its early stages, the targeted nature of phages offers a compelling alternative to traditional antibiotics. Continued research and clinical trials will be crucial in determining the most effective formulations, dosages, and application methods for incorporating phage therapy into mainstream acne treatment protocols.

## 8. Conclusions: Discussion and Future Directions

Acne management is at a critical juncture where the increasing understanding of the skin microbiome’s role in acne pathogenesis is reshaping therapeutic strategies. Nonetheless, while studies have begun to elucidate the roles of different microbial communities—bacteria, fungi, and viruses—in acne pathogenesis, more research is needed to understand the complex interactions between these organisms and their impact on skin health. As research into microbiome interactions deepens, it has become clear that maintaining microbial balance is crucial for long-term skin health and effective acne management. While traditional antibiotic-based treatments have been effective in controlling acne, they come with significant drawbacks, particularly the disruption of the skin microbiome and the growing concern of antibiotic resistance. In the future, personalized acne treatments based on an individual’s unique microbiome profile may become a reality, offering more targeted and sustainable solutions reducing the need for broad-spectrum antibiotics. Advances in metagenomics and bioinformatics will play a pivotal role in enabling clinicians to tailor treatments that not only target acne but also preserve or restore the skin’s natural microbial ecosystem. Furthermore, developing guidelines for the judicious use of antibiotics and fostering awareness about antibiotic resistance among both patients and healthcare professionals will be essential for mitigating the risks associated with overuse. Emerging treatments such as probiotics, phage therapy, and microbiome-modulating agents offer promising alternatives to conventional antibiotic therapies. Probiotics, both topical and oral, have shown potential in restoring skin microbial diversity and improving skin health by modulating inflammatory pathways and skin barrier function. Phage therapy, with its specificity in targeting pathogenic bacteria like *C. acnes*, represents a novel approach that avoids the broad-spectrum effects of antibiotics, reducing the risk of dysbiosis and resistance. However, the efficacy and safety of these microbiome-targeted treatments should be validated through larger clinical trials. These studies should also explore the long-term impact of these therapies on microbiome and overall skin health. In conclusion, while significant progress has been made in understanding the microbiome’s role in acne, the development of non-antibiotic treatments that support microbiome health marks an exciting and necessary shift in acne management. As research continues to advance, the future of acne therapy lies in innovative approaches that balance efficacy with microbiome preservation, offering patients safer and more sustainable treatment options.

## Figures and Tables

**Figure 1 ijms-25-11422-f001:**
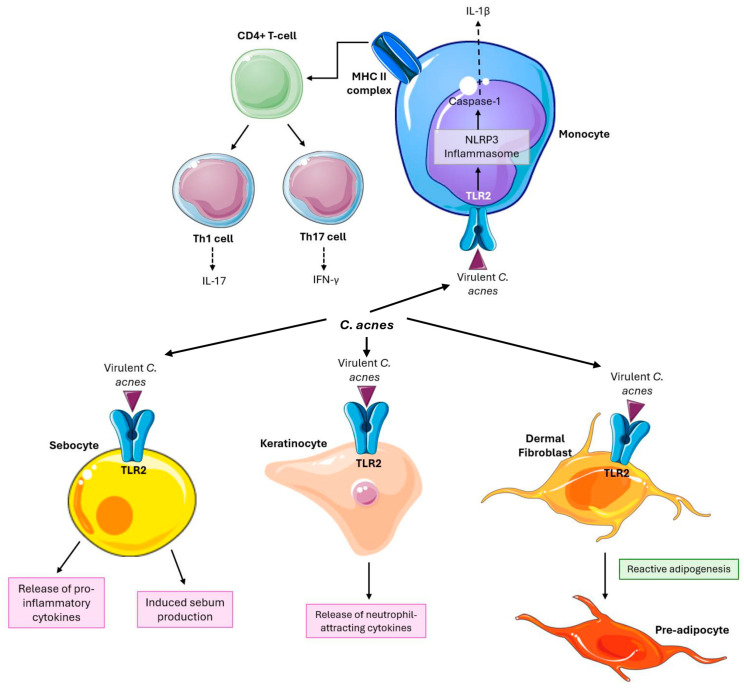
*C. acnes* plays multiple roles in acne development, including enhancing local inflammation and modulating immune responses. It stimulates sebocytes to release pro-inflammatory cytokines (TNFα, IL-6, IL-8, and IL-12) through TLR2 activation and induces IL-1β secretion via the NLRP3 inflammasome. *C. acnes* also drives Th17/Th1 responses, promoting IL-17A and IFNγ production. Additionally, *C. acnes* affects skin physiology by modulating keratinocyte differentiation lipid production and promoting reactive adipogenesis in dermal fibroblasts, contributing to acne pathogenesis.

**Table 1 ijms-25-11422-t001:** Key factors in acne development and progression [5].

Factors Contributing to Acne	Description
Microbiome alternations	Loss of *C. acnes* diversity; dominance of pathogenic strains (RT4, RT5) causing biofilm formation; increased virulence and resistance.
Immune system activation	Immune responses triggered (Toll-like receptors, Th17 pathway) leading to increased IL-1 and IL-17 cytokines.
Follicle changes	Genetic predisposition to hyperseborrhea, hyperkeratinization, and androgen sensitivity.
Environmental exposures	External factors like diet, stress, and pollutants (exposome) that exacerbate acne symptoms.

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
