# Peer review of "The Role of the Skin Microbiome in Acne: Challenges and Future Therapeutic Opportunities"

_ijms, 2024, doi:10.3390/ijms252111422_

Round 1
Reviewer 1 Report
Comments and Suggestions for Authors
Thanks for the authors for completion of this work. Although the idea of the manuscript is interesting and adds a lot to the scientific field, yet the article is too long and could be summarised especially in the introduction part. Beside many parts are repeated. The authors should revise the manuscript and try to present the data in more direct and simple way especially the introduction and conclusion.
Author Response
- Thanks for the authors for completion of this work. Although the idea of the manuscript is interesting and adds a lot to the scientific field, yet the article is too long and could be summarised especially in the introduction part. Beside many parts are repeated. The authors should revise the manuscript and try to present the data in more direct and simple way especially the introduction and conclusion.
We thank the reviewer for the comments. We have edited some parts of the review:
Introduction: line 31,32,36,38,42,44-57, added a table line 57 to simplify text and removed repetitive sentences 73-83. Other sections of the article were edited too line 116-133, 139-151; updated figure is now found line 214, 270-286; 296-303; 314-317; as shown in ijms-3257471 with corrections documents. Also, conclusions/discussion re-edited line 820-824; line 828-833. 844-846.
Please find corrected version in attachment with modifications

Reviewer 2 Report
Comments and Suggestions for Authors
This manuscript comprehensively reviews the evolving understanding of the skin microbiome's role in acne vulgaris. The authors effectively outline the complexity of microbial interactions on the skin, particularly focusing on Cutibacterium acnes (C. acnes). They also explore the potential of microbiome-targeted therapies, such as probiotics and bacteriophage treatments. The manuscript is well-structured, and the authors critically evaluate current therapies while discussing potential future directions in acne treatment.
Suggested Revisions:
-
In the "Composition of Skin Microbiota" section, since this review is focused on acne, the content from lines 83 to 156 should be rewritten to make it more relevant to the composition of the skin microbiota in the face and trunk, which are the primary areas affected by acne.
-
There is a missing subtitle for section 2.2, while section 2.3 has one. Please correct this inconsistency by adding a subtitle for section 2.2.
-
The section discussing bacteriocins provides a general overview of their antimicrobial properties, but it lacks specific information on how bacteriocins relate to acne pathogenesis. Please include additional content on the relevance of bacteriocins in acne treatment, particularly in targeting C. acnes and other acne-related pathogens.
-
Please provide additional evidence in line 337 regarding changes in the skin microbiota following the use of topical retinoids in acne patients, as this information is crucial for supporting the discussion.
-
The "Efficacy and Tolerability" section (3.3) appears unnecessary for this review and could be removed to streamline the content and maintain focus on microbiome-targeted therapies.
-
In addition to probiotics, please include content on prebiotics and synbiotics, as these are important components of microbiome-modulating therapies.
-
The conclusion mentions the role of viruses in acne pathogenesis, but there is no discussion of the mechanisms by which viruses contribute to acne in the main text. Please add relevant content on the role of viruses in acne pathogenesis in the body of the review to ensure coherence between the main text and the conclusion.
Author Response
- This manuscript comprehensively reviews the evolving understanding of the skin microbiome's role in acne vulgaris. The authors effectively outline the complexity of microbial interactions on the skin, particularly focusing on Cutibacterium acnes (C. acnes). They also explore the potential of microbiome-targeted therapies, such as probiotics and bacteriophage treatments. The manuscript is well-structured, and the authors critically evaluate current therapies while discussing potential future directions in acne treatment.
Comment1
Revisions suggested:
- In the "Composition of Skin Microbiota" section, since this review is focused on acne, the content from lines 83 to 156 should be rewritten to make it more relevant to the composition of the skin microbiota in the face and trunk, which are the primary areas affected by acne.
Response 1
We thank the reviewer, please find our comments:
Line 95: Section 2 – Title was changed from “Composition of the Skin Microbiota” to “Commensal and Pathogenic Dynamics in the Skin Microbiota”. The sentence relating to Malassezia, Aspergillus, Cryptococcus, Rhodotorula and Epicoccum fungi being found on the torso and feet was removed.
Comment2
There is a missing subtitle for section 2.2, while section 2.3 has one. Please correct this inconsistency by adding a subtitle for section 2.2.
Response 2
This has been corrected: Line 245 = section 2.2, Line 310 = Section 2.3
Comment3:
The section discussing bacteriocins provides a general overview of their antimicrobial properties, but it lacks specific information on how bacteriocins relate to acne pathogenesis. Please include additional content on the relevance of bacteriocins in acne treatment, particularly in targeting C. acnes and other acne-related pathogens.
Response3:
Ref 38 was added explaining the mechanism of how bacteriocins kill pathogens (starting from line 277). Additionally, Ref 39 (starting from line 296) was added demonstrating how bacteriocins isolated from human skin Staphylococcus are capable of specifically inhibiting C. acnes.
Comment 4:
Please provide additional evidence in line 337 regarding changes in skin microbiota following the use of topical retinoids in acne patients, as this information is crucial for supporting the discussion.
Response 4:
Additional references were added [90-92], line 584.
McCoy WH., Otchere E., Rosa BA., Martin J., Mann CM., Mitreva M. Skin ecology during sebaceous drought - How skin microbes respond to isotretinoin. J Invest Dermatol. 2019;139(3):732-735. 10.1016/j.jid.2018.09.023 [DOI] [PMC free article] [PubMed] [Google Scholar]
Kelhälä H-L., Aho VTE., Fyhrquist N. et al. Isotretinoin and lymecycline treatments modify the skin microbiota in acne. Exp Dermatol. 2018;27(1):30-36. 10.1111/exd.13397 [DOI] [PubMed] [Google Scholar]
Coughlin CC., Swink SM., Horwinski J. et al. The preadolescent acne microbiome: a prospective, randomized, pilot study investigating characterization and effects of acne therapy. Pediatr Dermatol. 2017;34(6):661-664. 10.1111/pde.13261 [DOI] [PubMed] [Google Scholar]
Comment 5:
The "Efficacy and Tolerability" section (3.3) appears unnecessary for this review and could be removed to streamline the content and maintain focus on microbiome-targeted therapies.
Response 5:
This was deleted (lines 427 to 440)
Comment 6:
In addition to probiotics, please include content on prebiotics and synbiotics, as these are important components of microbiome-modulating therapies.
Response 6:
Separate sections relating to prebiotics and synbiotics were added. Prebiotics – section 6.2 in line 745. Synbiotics – section 6.3 in line 766.
Comment 7:
The conclusion mentions the role of viruses in acne pathogenesis, but there is no discussion of the mechanisms by which viruses contribute to acne in the main text. Please add relevant content on the role of viruses in acne pathogenesis in the body of the review to ensure coherence between the main text and the conclusion.
Response 7:
In section 2 starting from line 111, more information about viruses was provided.
Please find corrected version in attachment with modifications

Reviewer 3 Report
Comments and Suggestions for Authors
Dear authors
Happy day.
The paper is fine but need some improvement. Kindly consider my comments.
Abstract and keywords
1- Abstract : Fin
2- Keywords: kindly add one or two more keywords.
The text body
3- in Introduction part kindly highlight the role of opportunistic pathogens in skin infection.
4- Did the microbes you focus on are opportunistic pathogens.
5- Did you think that “Composition of the Skin Microbiota “ is a correct title and if so did you cover all microbes under” Composition of the Skin Microbiota “. Kindly re-improve.
6- Kindly describe the role of cosmetics, disinfectants, perfumes (with alcohol) etc., in changing the skin microflora balance.
7- “2.3.2. Bacteriocins and Their Selective Inhibition of Pathogens” “2.3.3. Implications for Acne Treatment” concerning those two title kindly made them more related to 2.3.
8- I found it is logic to move “4. Antibiotic Use and Misuse in Acne Management “ to be before “3. Impact of Approved Non-Antibiotic Acne Treatments on the Skin Microbiome”
9- In section 4 you talk about antibiotic but in “4.1. Antimicrobial Resistance in Acne: Clinical Implications 430” you talk about the antimicrobial. It is not stylistic to do so. Kindly rearrange your idea.
with my pleasure

Author Response
Comment1
- Abstract and keywords
Abstract : Fine
Comment 1:
in Introduction part kindly highlight the role of opportunistic pathogens in skin infection.
Response 1:
We thank the reviewer for the valuable comment; please find two keywords: acne vulgarism, probiotics.
Comment 2
Did the microbes you focus on are opportunistic pathogens
Response 2
Yes, microbes like Cutibacterium acnes are considered opportunistic pathogens. They usually live on the skin without causing problems, but they can cause infections when conditions change, such as acne, we hope that this answer is satisfactory to your query.
Comment 3:
in Introduction part highlight the role of opportunistic pathogens in skin infection.
Response 3:
This was highlighted in the introduction line 41.
Comment 4:
Did you think that “Composition of the Skin Microbiota“ is a correct title and if so did you cover all microbes under” Composition of the Skin Microbiota“. re-improve.
Response 4
Changed to Commensal and Pathogenic Dymanics in the Skin Microbiota to highlight the interaction between harmless and harmful behaviors in microbial communities
Comment 5
describe the role of cosmetics, disinfectants, perfumes (with alcohol) etc., in changing the skin microflora balance.
Response 5
The following studies were added to the text relating to the ocean water exposure and cosmetics.
Nielsen, M. C., & Jiang, S. C. (2019). Alterations of the human skin microbiome after ocean water exposure. Marine pollution bulletin, 145, 595–603. https://doi.org/10.1016/j.marpolbul.2019.06.047
Lee H.J., Jeong S.E., Lee S., Kim S., Han H., Jeon C.O. Effects of Cosmetics on the Skin Microbiome of Facial Cheeks with Different Hydration Levels. Microbiologyopen. 2018;7:e00557. doi: 10.1002/mbo3.557.
Comment 6
“2.3.2. Bacteriocins and Their Selective Inhibition of Pathogens” “2.3.3. Implications for Acne Treatment” concerning those two titles made them more related to 2.3.
Response 6
- “Bacteriocins and Their Selective Inhibition of Pathogens” was changed to “Staphylococcal-Produced Bacteriocins and Their Selective Inhibition of Pathogens” in line 271 and “Implications for Acne Treatment” was changed to “The Potential of Staphylococcus in Acne Treatment” in line 290.
Comment 7
I found it is logic to move “4. Antibiotic Use and Misuse in Acne Management “ to be before “3. Impact of Approved Non-Antibiotic Acne Treatments on the Skin Microbiome”
Response 7
Thank you, done this was moved. Antibiotic Use and Misuse in Acne Management now section 3 line 443
Impact of Approved Non-Antibiotic Acne Treatments on the Skin Microbiome now in section 4 line 559.
Comment 8
In section 4 you talk about antibiotic but in “4.1. Antimicrobial Resistance in Acne: Clinical Implications 430” you talk about the antimicrobial. It is not stylistic to do so. rearrange your idea.
Response 8
The title has been changed to “Antimicrobial Resistance in Acne: Mechanisms, Complications and Recommendations”, with some changes in the text line 461-463. The sub-title:
“3.1 Clinical implications of antibiotic resistance in acne” has been moved further up.
We thank the reviewer for the valuable comments, we went through the pdf and edited the typo errors. Please find corrected version in attachment with modifications
